# Combination of GC-MS Molecular Networking and Larvicidal Effect against *Aedes aegypti* for the Discovery of Bioactive Substances in Commercial Essential Oils

**DOI:** 10.3390/molecules27051588

**Published:** 2022-02-28

**Authors:** Alan Cesar Pilon, Marcelo Del Grande, Maíra R. S. Silvério, Ricardo R. Silva, Lorena C. Albernaz, Paulo Cézar Vieira, João Luis Callegari Lopes, Laila S. Espindola, Norberto Peporine Lopes

**Affiliations:** 1Núcleo de Pesquisa em Produtos Naturais e Sintéticos (NPPNS), Departamento de Ciências Biomoleculares, Faculdade de Ciências Farmacêuticas de Ribeirão Preto, Universidade de São Paulo—USP, Ribeirão Preto 14040-903, Brazil; pilonac@gmail.com (A.C.P.); marcelo.grande@yahoo.com.br (M.D.G.); mairarosato@gmail.com (M.R.S.S.); rsilvabioinfo@gmail.com (R.R.S.); paulocezarv@gmail.com (P.C.V.); joaoluis@usp.br (J.L.C.L.); 2Laboratório de Farmacognosia, Campus Universitário Darcy Ribeiro, Universidade de Brasília, Brasília 70910-900, Brazil; lorena.albernaz@gmail.com (L.C.A.); darvenne@gmail.com (L.S.E.)

**Keywords:** dengue, essential oils, large datasets, molecular networking, larvicidal activity

## Abstract

Dengue is a neglected disease, present mainly in tropical countries, with more than 5.2 million cases reported in 2019. Vector control remains the most effective protective measure against dengue and other arboviruses. Synthetic insecticides based on organophosphates, pyrethroids, carbamates, neonicotinoids and oxadiazines are unattractive due to their high degree of toxicity to humans, animals and the environment. Conversely, natural-product-based larvicides/insecticides, such as essential oils, present high efficiency, low environmental toxicity and can be easily scaled up for industrial processes. However, essential oils are highly complex and require modern analytical and computational approaches to streamline the identification of bioactive substances. This study combined the GC-MS spectral similarity network approach with larvicidal assays as a new strategy for the discovery of potential bioactive substances in complex biological samples, enabling the systematic and simultaneous annotation of substances in 20 essential oils through LC_50_ larvicidal assays. This strategy allowed rapid intuitive discovery of distribution patterns between families and metabolic classes in clusters, and the prediction of larvicidal properties of acyclic monoterpene derivatives, including citral, neral, citronellal and citronellol, and their acetate forms (LC_50_ < 50 µg/mL).

## 1. Introduction

Dengue is a viral infection transmitted mainly by the female *Aedes aegypti* mosquito. This disease mainly affects tropical regions, depending on the rain precipitation rate, temperature, humidity and urbanization process [1,2,3,4,5,6,7]. The number of cases has increased almost 8-fold over the last 2 decades (from 505,430 in 2000 to 2.4 million in 2010; 4.2 million in 2019), leading to the death of more than 4000 people in 2015. Regions of Latin America, East Asia and the Western Pacific account for over 70% of the cases. In Brazil, more than 1.5 million cases (more than 1000 deaths) were recorded in 2016 alone [1,7,8].

The WHO indicates that combatting the transmitting mosquito is the most efficient strategy to control and prevent dengue [9]. However, there are still no specific insecticides or repellents (natural or synthetic) to exclusively combat the *Ae. Aegypti* mosquito. Consequently, insecticides and repellents can cause disturbances and/or the mortality of important insects, such as bees and ants, resulting in the degradation of important ecosystem services, including the pollination of crops [9]. Thus, nontoxic and specific repellent agents or larvicides against *Ae. aegypti* are urgently required to both reduce the number of arboviral disease cases (dengue, Zika and chikungunya) and protect the ecological roles of insects.

In order to address the aforementioned concerns, the Brazilian National Dengue Control Program has promoted the replacement of these synthetic compounds with substances that are less harmful to the environment. Organophosphates (malathion, fenitrothion and temephos) initially replaced organochlorines, which were in turn replaced by pyrethroids (cypermethrin and deltamethrin). However, these compounds still present some toxicity and continue to endanger populations of pollinating insects, animals and the environment [8].

Since the last century, efforts have been made to source products of natural origin with activity against *Ae. Aegypti*, such as pyrethrum and neem oil; although, no commercial products have been approved/authorized by Brazilian regulatory bodies to date. The literature reports the potential of various natural insecticides, especially those of microbial and plant origin [10,11]. The arms race between insects, plants and microorganisms constitutes a large part of the known metabolic arsenal, the so-called secondary metabolites or “special” metabolites. Today, it is estimated that more than 100,000 plant-derived metabolites may have some activity against insects and microorganisms [12].

Among the advantages of natural pesticides/larvicides, we can highlight their environmental safety, biodegradation and multiple mechanisms of action through synergistic effects. However, while the biodegradation of botanical active ingredients may sound positive, it actually represents a double-edged sword. There is generally greater compatibility between the released biocontrol agents and other natural enemies and greater safety for bees and other pollinators [8,13]. However, the lack of persistence of these bio-insecticides in crops under real field conditions brings some disadvantages. Most botanical insecticides are highly susceptible to photodegradation (e.g., pyrethrins), abiotic oxidation (azadiractins) or volatilization loss (essential oil terpenoids) when applied outside a controlled environment (e.g., indoors), requiring their reapplication when used on monocultures. Despite this limitation, certain botanical insecticides have proven records dating back 2–3 decades, confirming their effectiveness in the field [14,15]. Between 2007 and 2016, the state of California used azadirachtin, chenopodium and natural pyrethrins as the main botanical assets for pest control. However, recent formulation developments present opportunities to dramatically improve the field performance of botanical insecticides in terms of their efficacy and persistence [16,17].

The search for larvicidal agents present in essential oils (EO), plants or natural extracts requires increasingly modern analytical and computational tools, since these natural products are composed of dozens and even hundreds of compounds [18,19,20,21]. Recently, innovative methods based on LC-MS*^n^* have been applied for untargeted metabolomic analysis to accelerate the structural annotation of compounds [12,22,23,24,25]. The increased sharing of experimental MS/MS data and the growing number of spectral databases, such as NIST, METLIN, MassBank, MASST, NuBBE_DB_, Sumner/Bruker and ReSpect, have promoted the development of several bioinformatics approaches that help in the interpretation of large MS/MS datasets [26,27,28,29,30,31,32,33].

One of these approaches is the concept of spectral similarity networking (so-called molecular networks—MN) which is based on the organization and visualization of MS/MS data via spectral similarity (homologous fragments) [24,34,35,36,37,38,39]. Structurally related compounds often share similar MS/MS spectra. The MN groups these compounds (nodes) according to the degree of spectral similarity as a network in a knowledge graph format, thus allowing visual exploration of identical/analogous molecules and accelerating the identification of subgroups or characteristics of a given group of molecules. Chemical annotation via molecular networking arises from the combination of direct spectral correspondence between MS/MS spectra and compound libraries (MS/MS data) and through the relationship of molecular network masses (differences) between closely related structures (degree of spectral similarity). A mass difference of 15 Da between nodes with a high degree of similarity may suggest a CH_3_ group for the same class of compound, while differences of 162, 146 or 132 Da may correspond to homologues glycosylated with hexose, deoxyhexose or pentose [36,39,40,41,42].

In this study, we propose a strategy combining the spectral similarity networking (molecular networking) approach with larvicidal activity tests against *Ae. aegypti* to analyze commercial essential oils with the aim of discovering potential bioactive metabolic classes. The GC-MS retention time and fragmentation, chemotaxonomy and larvicidal activity against *Ae. aegypti* (LC_50_ values) of essential oils were organized, grouped and evaluated by molecular networking. Figure 1 shows a graphical representation of the strategy.

## 2. Results

Essential oils from 20 plant species (from 9 families) with insecticidal properties were analyzed by an untargeted profiling method using GC-EI/MS (Appendix A) and tested against *Ae. aegypti* larvae (third instar, L3), evaluating their mortality rate at 24 h and 48 h, Table 1. According to Dermaque et al. [43], an initial screening strategy to preselect an extract active against *Ae. aegypti* larvae involves testing at a single concentration, standardized at 250 ppm.

Species of the genera *Lavandula*, *Cymbopogon*, *Rosmarinus*, *Citrus*, *Perlagonium* and *Amyris* demonstrated larvicidal activity, while the other essential oils presented little or no activity. Among them, it is noted that the essential oils of the genus *Eucalyptus,* as well as the species of *Cymbopogon* and *Litsea*, had highly active LC_50_ values of: *Eucalyptus citriodora* (23.3 µg/mL), *Eucalyptus staigeriana* (43.1 µg/mL), *Cymbopogon nardus* (31.3 µg/mL), *Cymbopogon flexuosus* (41.7 µg/mL) and *Litsea cubeba* (32.7 µg/mL).

In order to annotate and streamline the process of discovering bioactive classes in essential oils, we applied the molecular network approach with retention time and fragmentation data from the GC-MS/MS experiments and metadata, such as taxonomy and LC_50_ values calculated in the *Ae. aegypti* larvicidal tests.

### Molecular Networking

The molecular network of 20 essential oils resulted in 82 nodes (with qualitative, quantitative and metadata for each compound), connected by 258 edges and grouped into a large cluster with 76 nodes and 6 lone pairs, as shown in Figure 2. One of the advantages of MN is the ability to create filters for nodes and edges in order to recognize patterns in the dataset. In this sense, different colors were attributed to nodes considering the retention time of the compounds present in the GC-MS profiles (red—longer RT; yellow—shorter RT) and to the relative abundance of ions represented by the node sizes (Figure 2).

The distribution of colors in the cluster regions indicated that different classes of metabolites must be grouped differently. In addition, most of the compounds (nodes) were eluted between 7 and 20 min, which may denote certain metabolic classes and, consequently, assist in the annotation process.

Another aspect is to evaluate the correlation of these compounds between families and their chemotaxonomic characteristics. It is also possible to make an indirect association between the chemotypes and the larvicidal potential of the cluster regions.

Figure 3 shows the MN using the compound distribution filter (relative abundance of ions) among the families using a color gradient in the nodes. Yellow represents little or no abundance, while blue represents a high abundance of ions. The Burseraceae and Cupressaceae families presented a strong correlation with most of their compounds (nodes) located in the region with the lowest retention times. Geraniaceae and Lamiaceae displayed a wide distribution of their compounds in the MN, while metabolites in Lauraceae, Myrtaceae, Poaceae and Rutaceae were observed between 7 and 20 min. Pinaceae concentrated its compounds in longer retention times.

Larvicidal activity against *Ae. aegypti* was used as a filter for the third step of evaluating the molecular network of commercial essential oils. The LC_50_ values (Table 1) from essential oils were used in two different ways to calculate the individual larvicidal activity for each node (substance). First, we calculated the average of the LC_50_ values for each node (one substance may be present in different EO), Figure 4A. Alternatively, we calculated the relative average considering ion abundance (present differently in each EO) for each node, Figure 4B. In both molecular networks (Figure 4A,B), we colored the maximum and minimum calculated LC_50_ values in four equidistant categories. The pink color represents the most active nodes, followed by blue, green and yellow. Although the ranges were different between the different averages, the color pattern was similar.

As a result, it was possible to observe three regions of molecular network with different larvicidal potentials. It is important to emphasize that these regions are projections of bioactivity, but may be used as a guide for regions/compounds to be explored and studied. In this case, for both averages (molecular networks) the lower right side of the cluster, colored in pink and blue, suggests a region with higher larvicidal potential. In Figure 4B, there is also a pink diagonal projecting this potential.

The fourth filter of this molecular networking was to annotate the nodes. We used the GNPS library combined with our in-house NIST database and gas-phase fragmentation data. The annotated compounds were then classified hierarchically according to the NP-Classifier ontology [44], as shown in Figure 5.

Figure 5 represents the filtered molecular network for compound annotation. The ellipse represents the level of the superclass (terpenes), while the colors represent the distribution of classes into monoterpenes (orange) and sesquiterpenes (purple). In the lower part of Figure 5, it is possible to observe the group of expanded monoterpenes and sesquiterpenes. Eight subclasses were found for monoterpenes: acyclic (orange), camphane (green), fatty alcohols (light blue), menthane (blue), monocyclic (purple), pinane (pink) and thujan (red), while six subclasses were classified as sesquiterpenes: cadinan (yellow), caryophyllane (green), elemane (light blue), germacrane (blue), himachalane (dark blue) and longifolene (pink) alcohols.

The potential bioactive nodes were annotated and classified as acyclic (orange), camphane (green) and menthane/monocyclic monoterpenes (dark blue). Specifically, the compounds with the lower LC_50_ values were annotated as citral (polyunsaturated and aldehyde as a functional group (FG)), neral (FG-aldehyde), citronellal (monoterpene-unsaturated and FG-aldehyde), citronellol (unsaturated and FG-alcohol), citronellol acetate (unsaturated and FG-acetate), isopulegol (unsaturated, monocyclic and FG-alcohol), linalool (poly-unsaturated and FG-alcohol), camphor (menthane and FG-ketone), and endo-borneol (menthane and FG-ketone).

Table 2 shows the list of annotated compounds, retention time (min), precursor ion values (*m*/*z*), samples and relative LC_50_ values.

To verify the predictive potential of the strategy to discover larvicides in essential oils, we tested some of the monoterpenes indicated by MN against *Ae. aegypti* larvae and calculated their LC_50_ values after 48 h exposure. Among the compounds tested were: α-fenchene (>100 µg/mL); eucalyptol (>100 µg/mL); menthol (>100 µg/mL); citronellol (65.3 µg/mL); citronellal (57.8 µg/mL); cymol (41.7 µg/mL); citral (40.1 µg/mL); α-phellandrene (40.1 µg/mL) and D-limonene (27.1 µg/mL).

Similarly, we projected the LC_50_ values of the tested compounds onto the molecular network to confirm the pharmacological patterns pointed out by MN, Figure 6. As a result, the diagonal indicated by MN including acyclic, monocyclic monoterpenes and menthane-type monoterpenes were active against *Ae. aegypti* larvae. In addition, some menthane-type monoterpene derivatives, such as D-limonene, were highly active.

## 3. Discussion

Essential oils and their structural analogues have historically made an important contribution as repellents or insecticides against *Ae. aegypti* in different communities [8,14,45,46]. However, larvicide/insecticide products based on natural products (NP) are scarce in the industry, revealing some difficulties associated with applying traditional approaches in NP [43,47,48].

Typically, a traditional approach uses organic solvents to produce crude extracts (polar or non-polar) that are screened for pharmacological activities and then fractionated into dozens of portions for further iterative bioguided analyses. In each cycle, the active fractions are reassembled after verification with spectral experiments, which can range from simple UV light to more expensive NMR analyses. Thus, the cost of pinpointing the NP-based product by traditional methods is high when considering that hundreds of extracts need to be analyzed to find a hit molecule [47,49].

This challenge has been partially addressed through the development of dereplication methods, which aim to prioritize the discovery of bioactive substances, still in the crude extract, avoiding the re-isolation of interferents or known compounds. Dereplication uses previously established data (spectroscopic data, pharmacological and physicochemical properties) present in databases, scientific literature and computational tools to compare samples and reference material and, therefore, annotate and to attribute some properties to substances still in extracts or fractions. Despite offering advantages over traditional methods, this strategy faces some obstacles, as follows: (1) the databases and literature are not comprehensive and standardized, rendering it difficult to discover non-ubiquitous compounds; (2) the comparison process is still univariate between the sample(s) and the reference material, making the whole process time-consuming; (3) the global relationships between samples and metabolites are little explored [18,25,50,51].

On the other hand, chemometrics and bioinformatics have faced these obstacles introducing a holistic view of how to manage data and especially how to extract relevant information from vast datasets. Some of the strategies employed in genomics and proteomics are gradually being introduced into NP science. One such strategy is the use of spectral similarity information (or any other molecular data) that can provide clues as to how the constituents of a sample are organized (classes, substituents and properties) [38,40,52,53,54].

This organization concept is already used in the GNPS platform generating spectral networks of MS/MS data. Therefore, not only can single samples be screened for similarity, but hundreds or even thousands of samples can be organized into a single network, the so-called molecular network. Thus, using a spectral similarity score (cosine score) we can organize families, classes, substituents from one or several samples at once and still discover distribution patterns of unknown metabolites [25,55,56,57,58,59].

However, several parameters need to be adjusted to extract relevant information from the MS/MS data. In LC-MS, MS/MS experiments are performed in data dependent analysis (DDA), which means that each ion is initially isolated from the others, fragmented and then analyzed (collected). In contrast, most GC-MS systems do not possess an ion isolation chamber and the separation of substances depends on chromatographic resolution, which is often insufficient to separate all of the metabolites. This results in the fragmentation of more than one substance at the same time, making it difficult to apply spectral comparison (similarity) tools between samples and the reference [53,60].

In this sense, our strategy largely overcomes these limitations, since the MS/MS spec-tral data from GC-MS are initially deconvoluted and aligned by the Mzmine 2 tool and then compared and organized by spectral similarity networking (molecular network GNPS). As shown in Figure 2, Figure 3, Figure 4, Figure 5 and Figure 6, it is possible to clearly visualize how substances are distributed throughout the chromatographic analysis, showing potential metabolic classes with different fragmentation profiles. It was possible to group information at the family level and discover patterns among them. We noticed that some families share similar MS/MS profiles, some of which have similar pharmacological properties. This allows us to extract chemotaxonomic information and prioritize bioactive families [61].

The incorporation of metadata into the molecular network opens up new opportunities to discover unknown patterns in samples. The relative mean of LC_50_ values from larvicidal experiments in *Ae. aegypti*, in several samples, allowed us to estimate the pharmacological effect of individual compounds that were repeatedly present in the samples and also indicate the bioactive ones. This is an innovative strategy in terms of discovering bioactive compounds in crude extracts, particularly in GC-MS experiments.

We annotated compounds quickly and easily using our in-house library combined with the molecular network. Using the unknown to known annotation principle, the compounds determined by the library were used in the network to evaluate the neighbors (similar) and their spectra, facilitating and speeding up the annotation. During this process we also noticed that pharmacological patterns were associated with some metabolic classes such as acyclic, monocyclic and methane monoterpenes. Furthermore, they rationally shared some characteristics such as the presence of oxygen in the form of an alcohol or carbonyl group (acetate, aldehyde or ketone), an unsaturation index between 1 and 3, and masses between 152 and 156 Da. This suggests that this strategy can be employed to indicate potentially promising chemical classes.

Finally, we tested the larvicidal activity of some of the isolated monoterpenes, revealing the bioactive potential of some of them. We therefore confirmed the potential of the strategy to not only predict the pharmacological activity of compounds in crude extracts and fractions, but also facilitate pattern recognition in samples and metabolites.

## 4. Materials and Methods

The essential oils were purchased from BioEssência^®^, Jaú, Brazil, and analyzed by dissolving in ethyl acetate at a concentration of 5 µg/mL. Table 1 details the percentages of the major compounds in each essential oil.

### 4.1. Larvicidal Activity against Ae. aegypti

Larvicidal tests were performed with the *Ae. aegypti* Rockefeller strain. Third instar larvae (L3) were obtained from infection-free colonies maintained in the insectary of the Laboratory of Pharmacognosy of the University of Brasília. Colony maintenance is in accordance with World Health Organization guidelines [62] Monthly monitoring of this strain, which is susceptible to insecticides, using dose–response curves performed in 12-well plates with 10 L3 larvae, with temephos as the positive control (concentrations ranging from 0.05 to 0.003125 µg/mL).

We optimized the WHO larvicidal trial to perform rapid screening and subsequent scale-up without harm. Assays were performed as described by Silva et al., 2020 [63], using 12-well plates, with 3 mL of tap water, 10 L3 larvae and 50 µL of sample diluted in DMSO. This test is rapid, uses a small sample and allows the screening of many essential oil samples for major compounds.

The samples were tested in quadruplicate using a negative control of 0.025% dimethyl sulfoxide in tap water at pH 7.75, conductivity at 34.5 µS/cm and temephos as the positive control. This organophosphate is used as a positive control due to its efficiency against *Ae. aegypti* Rockefeller strain (100% mortality at 0.35 µg/mL after 24 and 48 h with LC_50_ of 0.019 µg/mL), being used in private companies in Brazil as a pest control agent.

For initial screening, only the mortalities of essential oils at the final concentration of 250 µg/mL were determined, after 24 h and 48 h. The 50% lethal concentration values (LC_50_ µg/mL) were estimated using the test concentrations 250, 125, 62.5 and 31.25 µg/mL for the essential oils and 100, 50, 25, 12.5 and 6.25 µg/mL for pure compounds (GraphPad Prism 7.0 software, GraphPad, La Jolla, CA, USA). Larvae mortality was determined after a 24 h exposure treatment. For each bioassay, the temperature was maintained at 28 ± 2 °C and 70 ± 10% RH, with a 12 h photoperiod.

### 4.2. GC-MS Analysis

For the essential oil analysis, we used the analytical methodology described by Adams (2007) with adaptations. Analysis involved gas chromatography coupled with mass spectrometer detection (GCMS-QP2010) according to the following parameters: injector temperature, 250 °C; column temperature, 60 °C; heating ramp from 60 to 210 °C, at 3 °C/min, with a total time of 50 min; chromatographic column, DB-5, 30 m × 0.25 mm in diameter, 0.25 µm in thickness; helium was used as the carrier gas, under 79.7 kPa at 1.30 mL/min, with a linear velocity of 41.6 cm/s and 1 µL injection volume and a 1:60 split.

The mass detector parameters were as follows: ion source temperature, 250 °C; interface temperature, 260 °C; solvent cutoff at 3.0 min; Scan mode, from 35 to 400 *m*/*z*; detector voltage, 0 kV.

### 4.3. Molecular Networking

GC-EI/MS data were initially processed using the GCsolution (Shimadzu—Tokyo, Japan). Mass spectrometry molecular networks were created using the GNPS platform (http://gnps.ucsd.edu, accessed on 30 November 2021) [35,64]. As the mass data from the EI experiments did not present pre-selection of precursor ions (called acquisition format of DIA), a spectral deconvolution was necessary. To achieve this, GC-MS data were analyzed and processed using the MzMine 2 package according to the parameters shown in Table 3.

The files were submitted for processing by the spectral networks algorithm (GNPS) in three files: mgf file of the EI spectra deconvoluted by Mzmine 2, a quantification table of the peaks generated by Mzmine 2 and a metadata table, with information on the samples, such as LC_50_, taxonomy, coding, etc. In GNPS, the data were adjusted as follows: fragment ion mass tolerance of 0.5 Da; min matched peaks of 5; score threshold of 0.5. The advanced search options were: library class bronze; top history per spectrum of 1 and NIST and GNPS spectral libraries. In advanced network options: min pair cos 0.6 and network topK 10. For more details of the network on the GNPS, access: https://gnps.ucsd.edu/ProteoSAFe/status.jsp?task=e980401aaf22484f83adead45f6012dc, accessed on 30 November 2021.

Network visualization was performed in Cytoscape v.3.4.3. Node colors and sizes were mapped based on the metadata files, and the edge thickness attribute set to reflect cosine similarity scores, with thicker lines indicating greater similarity [65,66].

## Figures and Tables

**Figure 1 molecules-27-01588-f001:**
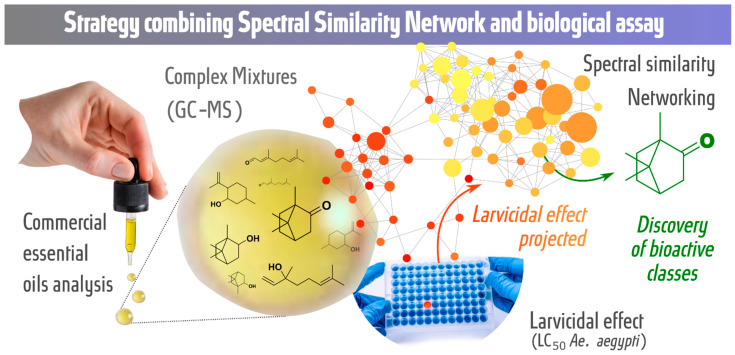
Strategy for the discovery of potential bioactive classes: spectral similarity networking (GC-MS data) combined with biological assays (larvicidal assay—LC_50_ values).

**Figure 2 molecules-27-01588-f002:**
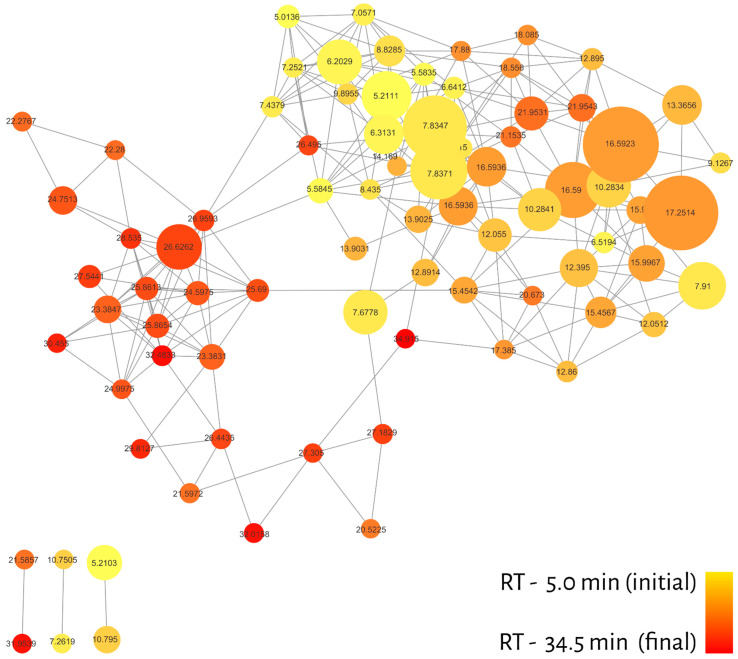
Molecular network filtered by retention time (RT). The color gradient is represented by yellow (initial) and red (end), while the node size represents the relative abundance of ions.

**Figure 3 molecules-27-01588-f003:**
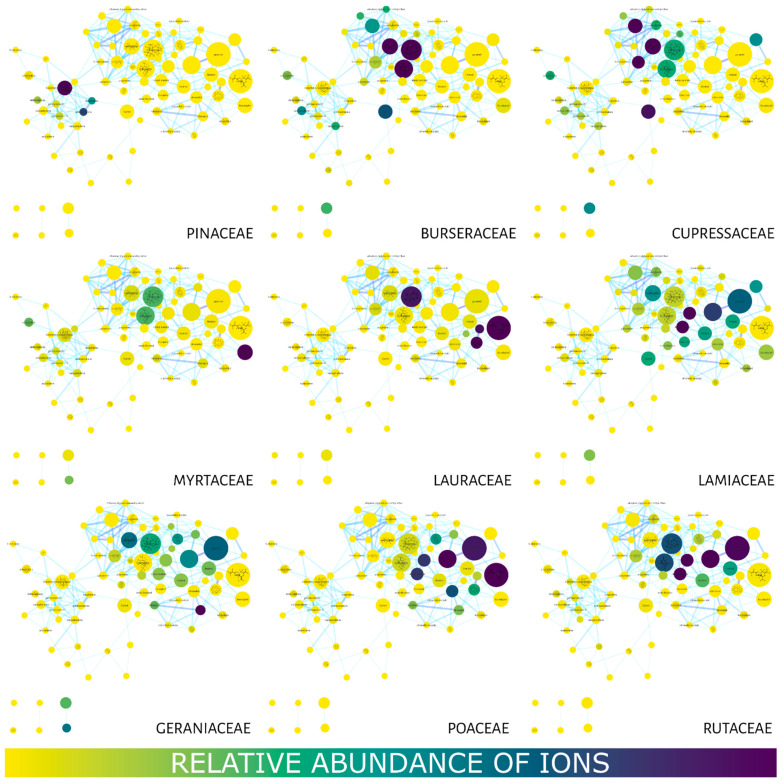
Molecular networks filtered for the relative abundance of ions present in the studied families. The color gradient represents ion (compound) abundance in the essential oils (family). Yellow—low or no abundance; blue—high abundance. The node size also represents the total relative abundance of ions.

**Figure 4 molecules-27-01588-f004:**
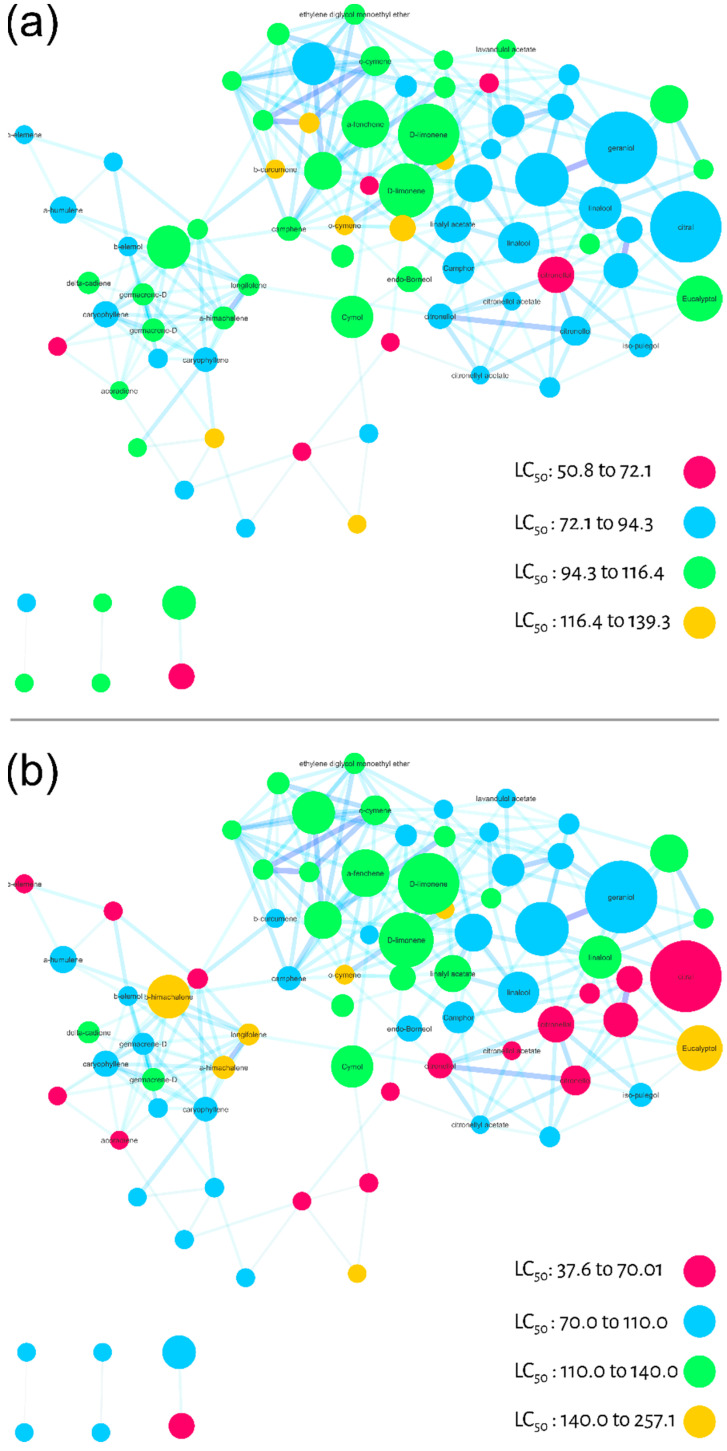
Molecular networks filtered by the calculated LC_50_ values (24 h larvae mortality) of the essential oils for each node. (**a**) Average of LC_50_ values calculated qualitatively (not considering the relative abundance of ions). (**b**) Relative average of LC_50_ values considering the relative abundance of ions.

**Figure 5 molecules-27-01588-f005:**
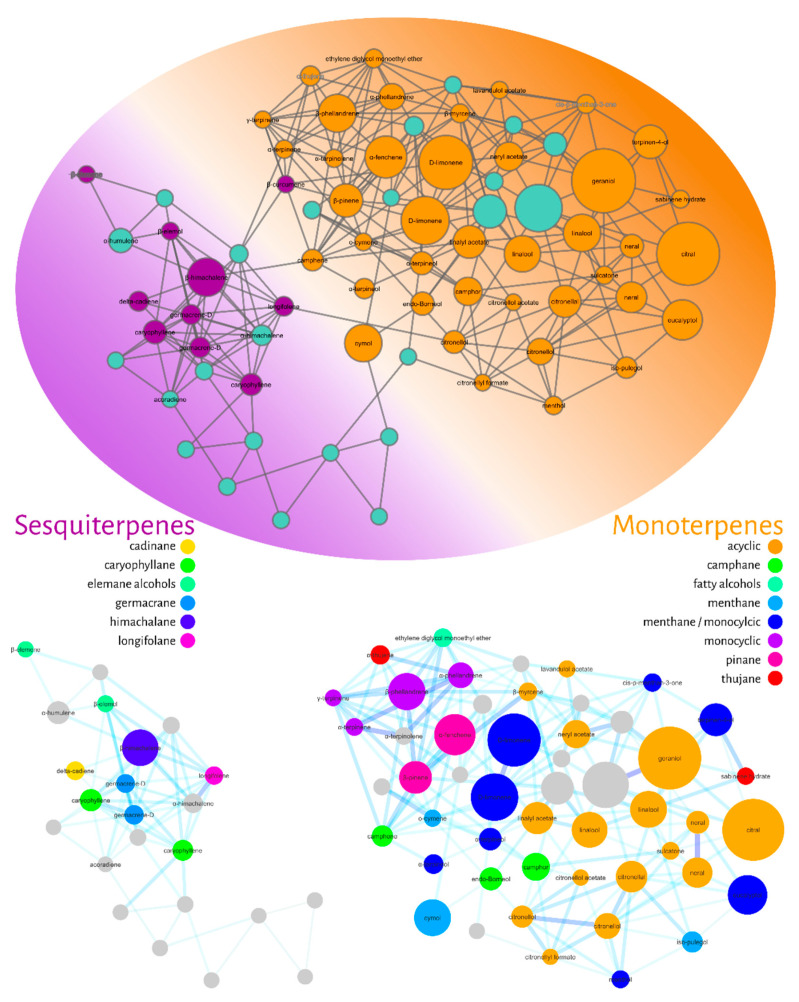
Molecular network filtered for compound annotation (nodes) using the GNPS library and an in-house NIST database. The annotated compounds were classified using the NP classifier ontology. The colors of the ellipse represent the annotated monoterpenes (orange) and sesquiterpenes (purple). At the bottom, the groupings of mono- and sesquiterpenes with their respective subclasses are expanded.

**Figure 6 molecules-27-01588-f006:**
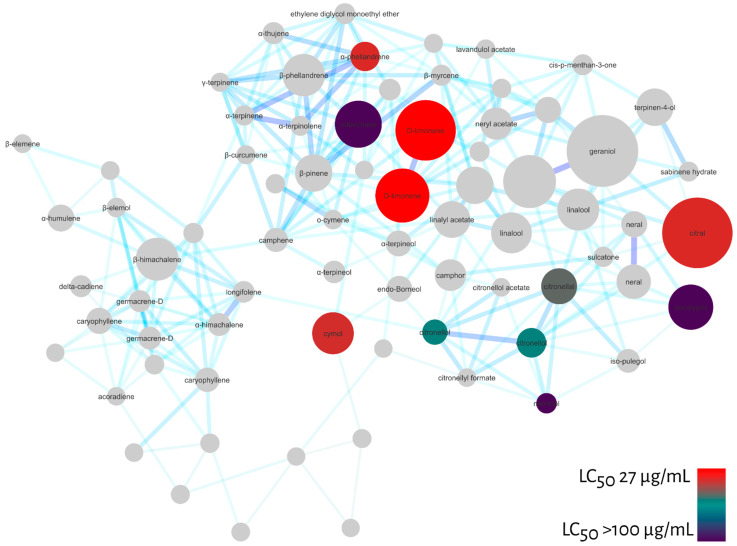
Molecular network filtered by LC_50_ values calculated for larval mortality at 48 h of *Ae. aegypti* for monoterpenes: α-fenchene (>100 µg/mL), eucalyptol (>100 µg/mL), menthol (>100 µg/mL), citronellol (65.3 µg/mL), citronellal (57.8 µg/mL), cymol (41.7 µg/mL), citral (40.1 µg/mL), α-phelandrene (40.1 µg/mL), D-limonene (27.1 µg/mL). Red nodes represent high larvicidal activity and dark blue nodules represent LC_50_ values > 100 µg/mL.

**Table 1 molecules-27-01588-t001:** Commercial essential oil larvicidal assay (mortality rate at 24 h and 48 h) results and corresponding LC_50_ values determined against *Ae. aegypti*.

Sample	Species (Family)	Batch	Major Compound (%)	Mortality 250 µg/mL (%, 24 h)	Mortality 250 µg/mL (%, 48 h)	LC_50_ (µg/mL) (24 h)
01	*Juniperus communis* (Cupressaceae)	180113	*α*-pinene (**38.9**)	75	82.5	135.2
02	*Origanum majorana* (Lamiacae)	180319	terpinen-4-ol (**25.2**)	82.5	80	121.3
03	*Cymbopogon martini* (Poaceae)	180227	geraniol (**80.6**)	87.5	92.5	73.88
07	*Boswellia carteri* (Burseraceae)	180217	*α*-pinene (**43.8**)	42.5	75	129.8
08	*Mentha piperita* (Lamiaceae)	180418	menthol (**45.7**)	100	100	95.29
09	*Citrus aurantium var. amara* (Rutaceae)	180206	D-limonene (**96.9**)	42.5	60	177.1
10	*Eucalyptus citriodora* (Myrtaceae)	180307	citronelal (**74.4**)	100	100	23.26
11	*Eucalyptus globulus* (Myrtaceae)	180205	eucalyptol (**89.9**)	87.5	97.5	276.6
14	*Lavandula angustifolia* (Lamiaceae)	180408	linalyl acetate (**63.0**)	100	100	85.88
16	*Lavandula hybrida* (Lamiaceae)	180403	linalool (**36.2**)	70	70	109
18	*Cymbopogon flexuosus* (Poaceae)	180326	citral (**50.6**)	100	100	41.66
19	*Cymbopogon nardus* (Poaceae)	180306	citronelal (**45.9**)	100	100	31.25
20	*Cedrus atlantica* (Pinaceae)	180226	*β*-himachalene (**54.7**)	60	65	269.1
21	*Rosmarinus officinalis* (Lamiaceae)	180415	camphor (**23.6**)	90	90	80.33
23	*Citrus aurantium subsp. Bergamia* (Rutaceae)	180402	D-limonene (**38.2**)	100	100	99.57
24	*Pelargonium graveolens* (Geraniaceae)	171234	citronellol (**35.3**)	100	100	78.32
27	*Litsea cubeba* (Lauraceae)	180412	citral (**47.7**)	100	100	32.74
31	*Salvia sclareia* (Lamiaceae)	180405	linalyl acetate (**71.0**)	60	75	120
33	*Amyris balsamifera* (Rutaceae)	180214	valencene (**21.5**)	100	100	99.51
34	*Eucalyptus staigeriana* (Myrtaceae)	180207	D-limonene (**29.2**)	100	100	43.13
N.C. ^1^	<1% DMSO	-		-	-	-
P. C. ^2^	Temephos (100% mortality)	-	-	0.35	0.35	0.019

^1^—negative control; ^2^—positive control.

**Table 2 molecules-27-01588-t002:** List of annotated compounds including the retention time (min), precursor ion values (*m*/*z*), samples and relative LC_50_ values.

RT (min)	Compound	*m*/*z* *	Samples	Relative LC_50_ ** (µg/mL)
5.01	thujene	93.1	1, 2, 9, 10, 11, 16, 18, 19, 23, 24, 31	125.6
5.21	pinene	91.1	1, 2, 7, 8, 9, 10, 11, 14, 16, 18, 19, 21, 23, 24, 27, 31	105.9
5.21	fenchene	93.1	1, 2, 7, 8, 9, 10, 11, 14, 16, 18, 19, 20, 21, 23, 24, 27, 31, 33	123.2
5.58	camphene	93.1	1, 2, 9, 10, 14, 16, 18, 19, 23, 27, 31	84.1
6.20	phellandrene	93.1	1, 2, 7, 9, 10, 11, 18, 20, 23, 24, 27, 31	128.5
6.31	pinene	93.1	1, 2, 8, 9, 10, 11, 14, 18, 23, 24, 27, 31, 33	116.0
6.52	sulcatone	43.0	1, 2, 3, 7, 8, 9, 10, 11, 14, 16, 18, 19, 20, 24, 27, 31, 33	55.1
6.64	myrcene	77.0	1, 2, 3, 9, 10, 11, 14, 18, 19, 23, 24, 31, 33	130.2
7.06	ethylene diglycol monoethyl ether	93.1	1, 2, 9, 10, 11, 14, 18, 19, 23, 24, 27, 31	127.5
7.25	terpinene	93.1	1, 9, 10, 11, 16, 18, 23	126.5
7.44	terpinene	136.1	1, 2, 9, 10, 23, 24, 27	120.3
7.68	cymol	119.1	1, 2, 3, 9, 10, 11, 14, 16, 18, 19, 20, 23, 24, 27, 31, 33	124.7
7.83	D-limonene	68.1	1, 2, 3, 8, 9, 10, 11, 14, 16, 18, 19, 20, 23, 24, 27, 31, 33	119.6
7.91	eucalyptol	43.0	1, 2, 3, 7, 8, 9, 10, 11, 14, 16, 18, 19, 20, 23, 24, 27, 31	229.6
8.44	cymene	93.1	1, 2, 3, 9, 10, 11, 14, 16, 18, 19, 24, 33	150.1
8.83	phellandrene	93.1	1, 2, 7, 9, 10, 11, 14, 18, 20, 23, 24, 27, 31	120.3
9.13	sabinene hydrate	71.1	2, 10, 18	120.9
9.90	terpinolene	93.1	1, 2, 9, 10, 11, 14, 18, 20, 23, 24, 27	124.0
10.28	linalool	71.1	1, 2, 3, 7, 9, 10, 11, 16, 18, 19, 20, 23, 24, 27, 31, 33	105.9
12.05	*iso*-pulegol	41.0	9, 10, 18, 19, 20, 23, 27, 31	86.5
12.06	camphor	95.1	9, 10, 16, 18, 20, 23, 27	86.8
12.40	citronellal	41.1	2, 10, 19, 20, 27, 31	51.9
12.86	menthol	112.1	10, 18, 27	93.0
12.89	endo-Borneol	95.1	1, 2, 9, 10, 16, 18, 19, 21, 23, 27, 33	94.8
12.90	cis-*p*-menthan-3-one	69.1	10, 16, 18, 23	96.7
13.37	terpinen-4-ol	71.1	1, 2, 7, 9, 10, 11, 16, 18, 31	121.5
13.90	terpineol	93.1	1, 2, 9, 10, 11, 14, 16, 18, 19, 20, 21, 23, 24, 27, 31, 33	125.2
15.45	citronellol	69.1	1, 2, 3, 7, 11, 18, 19, 20, 24, 27, 31, 33	48.3
15.98	neral	41.1	3, 10, 11, 18, 19, 20, 24, 27, 31	39.6
16.59	geraniol	69.1	2, 3, 7, 10, 11, 16, 18, 19, 20, 24, 27, 31, 33	97.2
16.59	linalyl acetate	93.1	2, 3, 7, 10, 11, 16, 18, 19, 20, 23, 24, 27, 31, 33	112.0
17.25	citral	69.1	3, 7, 10, 11, 16, 19, 20, 24, 27, 31, 33	37.6
17.39	citronellyl formate	109.1	10, 16, 24, 27	84.5
17.88	*unknown*	95.1	1, 2, 9, 16, 18, 23	92.9
18.09	lavandulol acetate	69.1	16, 18	104.5
18.56	*unknown*	69.1	3, 19, 20, 27, 33	78.2
20.52	*unknown*	119.1	1, 7, 9, 21	144.5
20.67	citronellol acetate	81.1	1, 20, 27, 31, 34	40.6
21.15	*unknown*	69.1	2, 3, 7, 11, 16, 18, 19, 24, 27, 33	114.3
21.59	*unknown*	41.0	1, 7, 9, 19, 24, 31, 33	76.4
21.60	*unknown*	119.1	1, 3, 7, 9, 10, 19, 27, 31, 33	92.6
21.95	neryl acetate	69.1	1, 2, 3, 7, 11, 16, 18, 19, 20, 24, 27, 31, 33	92.3
22.28	elemene	81.1	1, 7, 8, 9, 10, 20	58.7
23.38	caryophyllene	79.1	1, 2, 3, 7, 8, 9, 10, 11, 16, 18, 19, 23, 27, 31, 33, 34	97.8
24.60	himachalene	93.1	7, 8, 21, 34	247.8
24.75	humulene	93.1	1, 2, 3, 7, 8, 9, 10, 11, 18, 19, 20, 23, 27, 33, 34	79.1
25.00	acoradiene	93.1	1, 8, 9, 34	66.1
25.69	longifolene	93.1	1, 7, 8, 9, 20, 21, 27, 34	221.5
25.86	germacrene	91.1	1, 7, 8, 9, 10, 18, 19, 20, 21, 27, 33, 34	93.2
26.44	*unknown*	91.1	1, 2, 7, 8, 9, 10, 11, 21, 27, 34	100.9
26.50	curcumene	121.1	1, 2, 7, 9, 10, 11, 21, 33, 34	98.2
26.63	himachalene	119.1	1, 7, 8, 9, 16, 20, 21, 34	257.1
26.96	*unknown*	69.1	7, 8, 9, 18, 21, 24, 34	55.1
27.18	*unknown*	161.1	1, 7, 8, 9, 18, 19, 20	69.0
27.31	*unknown*	122.1	7, 8, 34	65.9
27.54	cadiene	119.1	1, 7, 8, 9, 10, 18, 19, 20, 21, 27, 34	128.1
28.54	elemol	107.1	9, 20, 34	91.4
29.81	*unknown*	91.1	1, 2, 3, 8, 9, 16, 18, 19, 21, 31, 33	100.1
30.46	*unknown*	95.1	7, 27, 34	38.6
31.95	*unknown*	91.1	1, 2, 7, 8, 9, 18, 20, 21, 34	103.8
32.02	*unknown*	161.1	1, 7, 9, 18, 20, 34	89.1
32.48	*unknown*	95.1	1, 7, 8, 9, 20, 34	83.9
34.92	*unknown*	69.1	3, 7, 34	60.9

*—precursor ion; **—relative average LC_50_ of each node.

**Table 3 molecules-27-01588-t003:** Mzmine 2 parameters for the commercial essential oil MS data analysis.

Feature	LVL 1	LVL 2	Value
Mass Detection	Scans		3.5–50.0 min
	Mass Detector		Centroid
		Noise Level	1.0 × 10^3^
ADAP Chrom. Build	Min. group size in # of scans		15
	Group intensity threshold		1.0 × 10^3^
	Min. highest intensity		1.0 × 10^3^
	*m*/*z* tolerance		0.01 *m*/*z*
Chrom. deconv.	Wavelets (ADAP)	S/N threshold	7
		S/N estimator	Intensity window SN
		Min feature height	1
		Coef./area threshold	30
		Peak duration	1.00
		RT wavelet range	0.15
	*m*/*z* center calculation	Median	
Spec. Deconv.	Multivariate Curve Resolution	Deconvolution window width (min)	0.15
		Retention time tolerance (min)	0.02
		Minimum number of peaks	1
ADAP Aligner	Min confidence (0 to 1)		0.05
	Retention time tolerance		0.1 (min)
	*m*/*z* tolerance		0.1 (*m*/*z*)
	Score threshold (0 to 1)		0.75
	Score weight (0 to 1)		0.1
	Retention time similarity	Cross-correlation	
Gap filling	Peak finder multithreaded		
		Intensity tolerance	0.1%
		*m*/*z* tolerance	0.2 *m*/*z*
		retention time tolerance	0.1 min

## Data Availability

Not applicable.

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
