# Peer review of "Combination of GC-MS Molecular Networking and Larvicidal Effect against Aedes aegypti for the Discovery of Bioactive Substances in Commercial Essential Oils"

_molecules, 2022, doi:10.3390/molecules27051588_

Round 1

Reviewer 1 Report

The authors should consider the followings:

  1. The authors should briefly describe the assay method optimization in each assay.
  2. Please give details on how the authors authenticate and continuous monitoring (and the frequency of monitoring checks) of the strain, Ae. aegypti.
  3. Did the authors perform tests across different batches of essential oils, species/family?
  4. Please provide the authentication details, the purity percentage, and batch number, of each essential oils were purchased from BioEssência in Table 1.
  5. Please explain why Tap water was used in 4.1 Methodology. And please provide the specification of the Tap water.
  6. The authors should clearly mention the novel findings of the research, in the part of abstract and conclusion.
  7. As in table 1, the authors should give better rationale of using 250 ug/ml as the cut-off value for evaluation.
  8. As in methodology 4.2, please give rationale(s) for choosing the scan mode, 35 to 400 m/z.
  9. The authors may consider adding a figure in summarizing the findings and explaining the workflow of the research (similar to a graphical abstract).
  10. The author may briefly expand or further elaborate in the part of Discussion.
  11. Please provide rationale(s) for choosing Temephos with its dosage used, as the positive control
  12. Please specific the vehicle control used. As for example, were all the sample dissolved equally, compared with the same percentage of DMSO used, across the functional assays.
  13. The authors should label the major peaks of Figure 1.
  14. In Figure 3 to 5, please enlarge the sub-figures to better visualize the details.

Author Response

Dear Reviewer 1,

We hope we were able to respond and correct all comments.
Thanks again for your attention

  1. The authors should briefly describe the assay method optimization in each assay.

We thank this recommendation and expanded our methodology in the text, as follow:

“We optimized the WHO larvicidal assay to perform rapid screenings and then scale up without harm. The optimized assay in 12-well plates is described in Silva et al., 2020 (Molecules. 2020 Aug 31;25(17):3978. doi: 10.3390/molecules25173978). Briefly, we used 12-well plates, with 3 mL of tap water, 10 L3 larvae and 50 uL of sample diluted in DMSO. This test is fast, uses a small sample and allows us to screen many samples of essential oils and major compounds.”

  1. Please give details on how the authors authenticate and continuous monitoring (and the frequency of monitoring checks) of the strain, Ae. aegypti.

We appreciated the concerns of the reviewer and provided the methodology used in our group.

“Monthly monitoring of Rockefeller strain, which is susceptible to insecticides, is carried out in the insectary of the Laboratory of Pharmacognosy at University of Brasilia (UnB), using dose-response curves performed in 12-well plates with 10 L3 larvae, with temephos as the positive control (concentrations ranging from 0.05 to 0.003125 µg/mL).”

  1. Did the authors perform tests across different batches of essential oils, species/family?

We do not test different batches of essential oils. However, we qualitatively analysed the chemical profile of each essential oil by GC-MS. This allowed us to correlate OE with the compounds responsible for the biological activity in question.

4.Please provide the authentication details, the purity percentage, and batch number, of each essential oils were purchased from BioEssência in Table 1.

Thank you for asking. As essential oils are complex mixtures, we have dosed the major compound for each EO and introduced in Table 1. We also include the batch number as requested.

  1. Please explain why Tap water was used in 4.1 Methodology. And please provide the specification of the Tap water.

We sincerely appreciate your question. Tap water was used because Guidelines For Laboratory And Field Testing Of Mosquito Larvicides (WHO, 2005) recommend it. The tap water specification is pH 7.75 and water conductivity 34.5µS/cm. This reference was introduced in the manuscript.

  1. The authors should clearly mention the novel findings of the research, in the part of abstract and conclusion.

We understand that a better explanation of our strategy should be provided. Therefore, part of the abstract was modified, as well as the discussion (conclusion) in an attempt to clarify the main objectives of the manuscript. We hope that such modifications meet your expectations.

  1. As in table 1, the authors should give better rationale of using 250 ug/ml as the cut-off value for evaluation.

We appreciate this important question. However, there is no description of concentration values in the Guidelines for Laboratory and Field Testing of Mosquito Larvicides (WHO, 2005), as this depends on the type of sample and its preparation. Recently, our research group published a work evaluating initial values for screening studies of extracts with potential activity against Ae. aegypti larvae. We found that tests at 250 ppm may be indicative (Front. Chem., November 15, 2021; doi.org/10.3389/fchem.2021.779049). We also included in text an explanation about the cut-off value.

  1. As in methodology 4.2, please give rationale(s) for choosing the scan mode, 35 to 400 m/z.

Thank you for asking. Essential oils are generally composed of terpenes, resulting from isoprenes (5 carbons), monoterpenes (10 carbons) and eventually sesquiterpenes (15 carbons). These classes have masses below 400 Da, taking into account the possibly oxidized derivatives. Thus, the range between 35-400 m/z was sufficient for these experiments. Furthermore, under electronic ionization at 70eV, molecular ions are barely observable and, narrow spectral ranges provide greater accuracy in spectral quality and, therefore, offer better results in the comparison process with reference libraries.

  1. The authors may consider adding a figure in summarizing the findings and explaining the workflow of the research.

We have replaced figure 1 (now in supplementary material) with the workflow of the research as suggested.

  1. The author may briefly expand or further elaborate in the part of Discussion.

The discussion section was reorganized to clarify the potential of the strategy. Thus, the entire text was revised and expanded as suggested.

  1. Please provide rationale(s) for choosing Temephos with its dosage used, as the positive control.

Temephos is commonly used as a positive control in larval assays for Aedes aegypti in susceptible strains (Rockefeller). In addition, temephos is still used in some situations, for example, private companies in Brazil that carry out pest control.

The result of the larvicidal activity of temephos was 100% larval mortality at 0.35 µg/mL at 24 h and 48 h and the LC50 of 0.0.019 µg/mL at 48h. Table 1 was also corrected and completed with mortality values of 100% at 24h and 48h.

  1. Please specific the vehicle control used. As for example, were all the sample dissolved equally, compared with the same percentage of DMSO used, across the functional assays.

The Guidelines for Laboratory and Field Testing of Mosquito Larvicides (WHO, 2005) describe that the most suitable solvent can be used for solubilization of the sample. In this case, we chose DMSO (< 1%) because it was able to solubilize all essential oils and is a solvent commonly used in biological studies, such as drug screening settings and biomedical applications.

  1. The authors should label the major peaks of Figure 1. (Alan)

Figure 1 was transferred to Supplementary material as suggested by reviewer 2.

  1. In Figure 3 to 5, please enlarge the sub-figures to better visualize the details.

All figures were enlarged to a better visualization and comprehension.

Reviewer 2 Report

Please find my comments below:

  • The introduction part is too long. Lines 81 to 110 can be deleted - they are not needed in this chapter.
  • Figure 1. is completely unnecessary. Instead, I would prefer a table with the percentage of compounds that make up the oils. This will make it easier to link the ingredients with possible biological properties.
  • By the way, please indicate in the methodology how the qualitative and quantitative composition of essential oils was fixed.
  • Please correct table 1. It is unreadable.
  • Pictures 2, 3, 4, 5. I don't know what they show because they are too small and also unreadable.
  • Did the authors have a chiral chromatography column if they managed to identify the enantiomer?
  • Table 2: Many compounds have two different retention times, why?
  • There is no proper discussion of the results with reference to already published works. Half a page is definitely not enough.
  • And please add a Conclusion chapter.

Author Response

Dear Reviewer 2,

We hope we were able to respond and correct all comments.
Thanks again for your attention

  1. The introduction part is too long. Lines 81 to 110 can be deleted - they are not needed in this chapter.

Thank you for recommendation. The introduction was reorganized and the aforementioned part was partially removed as suggested.

  1. Figure 1. is completely unnecessary. Instead, I would prefer a table with the percentage of compounds that make up the oils. This will make it easier to link the ingredients with possible biological properties.

Figure 1 has been replaced by a research object workflow. The data from Figure 1 was transferred to the supplementary material. The major compounds (in percentage) for each essential oil are shown in table 1.

  1. By the way, please indicate in the methodology how the qualitative and quantitative composition of essential oils was fixed.

For the analysis of essential oils, we used the analytical methodology described by Prof. Robert P. Adams, with the necessary adaptations for our sample and equipment, such as carrier gas flow, linear velocity and parameters in the mass detector. The reference of. Adams R. P., 1995 (Identification of Essential Oil Components by Gas Chromatography-Mass Spectroscopy, Allured Publ. Corp., Carol Stream, Illinois) was included in the material and methods section.

  1. Please correct table 1. It is unreadable.

Table 1 was corrected. We hope the modifications meet your expectations.

  1. Pictures 2, 3, 4, 5. I don't know what they show because they are too small and also unreadable.

Figures was corrected and enlarged. We hope the modifications meet your expectations

  1. Did the authors have a chiral chromatography column if they managed to identify the enantiomer?

Thank you for asking. We did not use chiral columns in our study. However, some enantiomers can be easily annotated due to the large differences in their retention times, such as α- and β-pinene. However, we agree that many of them can be difficult to separate and identify. Therefore, we decided to remove all stereochemical labelling, as raised by the reviewer.

  1. Table 2: Many compounds have two different retention times, why?

This is an important question and we appreciate your question. First, as raised by reviewer, different stereoisomers can be present in EO with different retention times and therefore multiple substances. Furthermore, during the pre-processing steps in Mzmine, eventually the algorithm may fail during in alignment process splitting a peak into 2 signals with very similar retention time (difference in seconds). For these cases, we analysed the signals that were divided and merged them into the most abundant ones. In addition, some signs were not annotated, but had important pharmacological results. We decided to keep them in the table including in the name description as unknown.

  1. There is no proper discussion of the results with reference to already published works. Half a page is definitely not enough.

The discussion section was reorganized to clarify the potential of the strategy. Thus, the entire text was revised and expanded as suggested.

  1. And please add a Conclusion chapter.

We understand your concern about the Conclusion section. However, the Molecules only suggests a discussion section, including a brief conclusion. So, in our correction, we have included a brief conclusion in the discussion section. We hope it can meet your expectations.

Round 2

Reviewer 2 Report

I don't have more questions